# Working with Disaster-Affected Communities to Envision Healthier Futures: A Trauma-Informed Approach to Post-Disaster Recovery Planning

**DOI:** 10.3390/ijerph19031723

**Published:** 2022-02-02

**Authors:** Heather Rosenberg, Nicole A. Errett, David P. Eisenman

**Affiliations:** 1Arup, Los Angeles, CA 90017, USA; 2Department of Environmental and Occupational Health Sciences, School of Public Health, University of Washington, Seattle, WA 98105, USA; nerrett@uw.edu; 3Division of General Internal Medicine and Health Services Research, David Geffen School of Medicine, University of California Los Angeles, Los Angeles, CA 90095, USA; deisenman@mednet.ucla.edu; 4Center for Public Health and Disasters, Department of Community Health Sciences, Fielding School of Public Health, University of California Los Angeles, Los Angeles, CA 90095, USA

**Keywords:** disaster, recovery, planning, trauma-informed

## Abstract

Disasters are becoming increasingly common and devastating, requiring extensive reconstruction and recovery efforts. At the same time, the level of available resources and the need to rebuild can present opportunities for more resilient land use and infrastructure, and to build healthier, more equitable and sustainable communities. However, disaster-affected individuals may experience trauma and mental health impacts that impede their ability to engage in long-range recovery planning. It is essential to consider and address community trauma when engaging with disaster-affected communities and in developing plans for recovery. Planners and engineers from outside the community (including public, private and non-profit practitioners) are often brought in to support long-term recovery. Most of these practitioners (particularly those focused on longer-range recovery) have no training in how disasters can affect mental health or what this could mean for their interactions with individuals or communities. In order to acknowledge and address disaster trauma in community recovery and redevelopment, we propose a trauma-informed approach which aims to provide practitioners supporting post-disaster community recovery planning guidance, in order to: avoid the causation of harm by re-traumatizing communities; better understand community needs; make sense of observed behaviors and avoid potential roadblocks; avoid becoming traumatized themselves; and facilitate community healing.

## 1. Introduction

Widespread devastation combined with unprecedented flows of external resources to support disaster recovery may present opportunities to “build back better [1].” Disaster response and recovery resources, including governmental and private sources, often represent an influx of money and technical support for affected communities. In the United States, major disaster declarations made under the Robert T. Stafford Disaster Relief and Emergency Assistance Act (the Stafford Act), authorize the provision of federal financial assistance for a wide range of public infrastructure and individual needs, including both emergency and permanent work [2,3,4]. Foundations, nonprofits and private donors may provide funds and human capital—including subject matter expertise and a boots-on-the-ground workforce—to redesign and reconstruct buildings and infrastructure.

With forethought, recovery investments can address pre-disaster vulnerabilities, inequities, and injustices by making structural improvements through redevelopment [5,6,7] and build social cohesion [8]. For example, the use of recovery resources to work towards goals outlined in the community’s comprehensive plan, versus simply restoring infrastructure and systems to their pre-disaster state, can help communities create economic and social value and become more resilient to future threats. 

As such, creating more resilient communities following disasters involves substantial long-range, pre-disaster, transformational recovery planning efforts, underpinned by community engagement. Decisions about rebuilding in a more resilient way require the engagement of community members to imagine a different future and to make decisions that may feel less familiar and more abstract. Diverse stakeholder engagement in recovery planning can help identify and prioritize community needs and resources, and help stakeholders align around a shared vision [5,8]. Multiple resources have been developed to inform community engagement in such planning, such as the NIST Community Resilience Planning Guide and the American Planning Association’s guidance for public engagement processes for disaster recovery [9,10].

Few local jurisdictions have developed recovery plans prior to a disaster [8]. In response, local government planners often work with external groups, including from federal and state government and for-profit consulting firms brought in following an event, to support long-range recovery planning [11]. These “post-disaster planning professionals” work to help restore a wide range of community infrastructure (e.g., transportation, energy and water infrastructure) and buildings (e.g., housing, businesses, civic buildings), as well as zoning considerations and other local policies (e.g., around land use). We focus here on those involved in longer range recovery, because they are least likely to have an awareness of the role of trauma in recovery efforts.

## 2. Challenges with Post-Disaster Recovery Planning

Following multiple engagements with communities in California, U.S.A., recovering from wildfires during the 2018 and 2019 fire seasons, one of the authors, H.R., and colleagues from Arup (a consulting firm that provides planning expertise to government clients) noticed recurring expressions of trauma among local government staff and community members. Examples included the need to share personal stories of the fires at the beginning of every meeting, challenges with decision makers who had been personally affected, and outbursts of anger from community members toward government officials and consultants in public forums. These experiences prompted a series of formative interviews, including six with state, local, and federal government officials from both within and outside of impacted communities, and two with planning professionals from other firms with experience supporting long term recovery efforts, to understand practitioners’ roles in engaging with communities, obstacles that they experienced, and encounters they had with community trauma. Respondents relayed stories from residents who had invested their entire savings in rebuilding their homes exactly as they had been before the disaster, only later to realize that they had not taken the steps to harden or adapt their property. As such, their homes were left vulnerable, and indeed, less valuable and insurable. This exemplifies the challenges of balancing community desires to quickly “return to normal” with the reality that previous conditions may no longer be able to withstand evolving hazard risks associated with a changing climate. 

Even with the best intentions, there are contexts where post-disaster recovery planning can face challenges. For example, planners from local communities that we interviewed shared that, in their experience, external groups may over-promise in order to ease local concerns, creating additional problems when they cannot deliver due to capability, scope or budget constraints. Outside groups may enter communities following disasters and position themselves as experts there to provide information and resources on communities, often underestimating the need for community story-sharing and the importance of integrating local knowledge and values in redevelopment. As a result, these post-disaster practitioners may become frustrated with the lack of progress on forward-looking decision-making while the community becomes frustrated that they are not being heard. This dynamic of expert/savior and community member/victim can perpetuate histories of marginalization, victimization and disempowerment and exacerbate feelings of distrust among the community toward government and authority. 

Perhaps most importantly, post-disaster planning and engineering practitioners from outside of the affected area may not have experience with historical traumatic events and how these have been compounded or exacerbated by the most recent devastating event. Appreciation of the determinants and impacts of trauma are missing from many approaches to community engagement in recovery processes. In fact, disciplines that focus on restoring the built environment (planners, engineers, architects and public administrators) rarely receive any training on mental health or the ways that trauma may influence the communities where they work. None of the individuals we interviewed had received any training in mental health. All expressed that such training would be helpful to them in their work.

## 3. Trauma and Disasters

According to the U.S. Substance Abuse and Mental Health Services Agency (SAMHSA), trauma is an event or series of events experienced as harmful or life-threatening that has persistent adverse effects on individual “functioning, and mental, physical, social, emotional, or spiritual well-being” [12]. Traumatized persons working on their community’s recovery may be simultaneously dealing with their own feelings of depression, anxiety, hopelessness and the resulting impairment to concentration and ability to work. Studies have found multiple patterns of people’s responses. Although most people will only experience transient distress that resolves on its own, up to 30% may experience PTSD early on, and others may develop trauma symptoms one year or more after the disaster [13]. The development of symptoms varies by the type of event, individual-level characteristics (age, access to resources) and biopsychosocial considerations [14]. 

Notably, the impacts of disasters, including trauma, may not occur equally or equitably, and may amplify pre-existing vulnerabilities and inequities [15,16]. Low-income populations, communities of color, individuals with disabilities, elderly people, immigrants and rural communities often live in places vulnerable to fire, flooding or other hazards. They have fewer resources and savings to rebuild, may be more dependent on under-resourced public services, and may have fewer opportunities to relocate. These same factors can also contribute to preexisting trauma that can be compounded by disaster. 

To cope with the stressful feelings brought on by recalling trauma, people may adopt avoidance behavior, i.e., avoiding reminders of the event or triggers such as limiting attendance at meetings or discussing the disaster. When meetings become intensely emotional, some persons will stop attending such gatherings, withdraw from participation in community recovery activities, and their voices and ideas, resources and concerns will not be heard [17]. Trauma and loss can create a significant sense of uncertainty, as well as anger and frustration with authority [18]. This can lead to anger or aggression, which may be expressed toward practitioners or in public fora. Such outbursts can be particularly challenging for practitioners and public agency staff who do not have training in mental health or conflict management. As such, processes designed to engage communities without considering their trauma may not be able to realize their intended goals, and may in fact become contentious and retraumatizing events. Conversely, processes that provide space for communities to acknowledge and work through trauma collectively can potentially enable better planning outcomes, and at the same time, support community healing.

## 4. What Is a Trauma-Informed Approach?

Trauma-informed practices include the following understandings: (a) trauma has widespread impact and there are various paths to recovery; (b) an individual’s experience of trauma may be accompanied with specific signs and symptoms of trauma; (c) the response from individuals, programs, organizations, and systems to the person experiencing trauma should be comprehensive and integrative; and (d) attempts should be made to prevent re-traumatization. Trauma-informed practices prioritize trust-building, transparency and social connectedness to reduce the impact of trauma through collaborative community processes [19]. Community members become seen as active participants in the process, rather than victims without control of their future. 

Trauma-informed systems seek to provide safe and supportive environments for clients and staff by incorporating best practices about trauma and its effects [20,21]. Although one purpose of these systems is to address traumatic reactions among individuals receiving trauma-specific services, they all seek to prevent re-traumatization for individuals with trauma exposure and to reduce secondary traumatic reactions among staff and family members [19,21,22].

In recent decades, mental health practitioners have adapted and implemented trauma-informed practices for a wide range of fields and professions, including emergency response workers, police, judicial, and other aspects of the criminal justice system. Less commonly, trauma-informed practices address the population level consequences of trauma and its aftermath [23,24]. However, examples of its use and impact exist. For instance, architects have used the concept of trauma-informed design to guide design of physical spaces [25]. In addition, the Centers for Disease Control and Prevention’s Office of Public Health Preparedness and Response (OPHPR) and SAMSHA developed a training for OPHPR staff on trauma-informed care and on the role of trauma in the communities they serve [26]. However, we are unaware of any resources that exist for post-disaster recovery planning professionals. 

## 5. A Trauma-Informed Approach to Recovery Planning

Trauma can adversely impact a community’s decision making; however, at present, there are no recommendations to help guide post-disaster recovery planning professionals to integrate, address or adapt to trauma while engaging communities in their work. In response, and based on our experience working with disaster-impacted communities in recovery, we propose a trauma-informed approach to post-event community-engaged disaster recovery planning.

A trauma-informed approach to recovery is not a step-by-step process for conducting community-engaged recovery work. Rather, we propose several recommendations aligned with each of the six trauma-informed principles (Table 1). The purpose of these recommendations is to help those supporting post-disaster recovery planning to: (a)First, do no harm and avoid re-traumatizing communities;(b)Better understand community needs;(c)Make sense of observed behaviors and avoid potential roadblocks;(d)Avoid becoming traumatized themselves;(e)Facilitate community healing;(f)Assist communities in choosing recovery pathways that enable them to “bounce forward” and leave them less vulnerable to future disasters.

We apply the definition of each of six principles that guide a trauma-informed approach (*Cultural, historical and gender issues; Safety; Trustworthiness and transparency; Peer support; Empowerment and choice; and Collaboration and mutuality*) to the context of trauma-informed recovery planning. We propose fourteen recommendations, each aligned with the trauma-informed recovery principles (Figure 1), to guide planning professionals’ implementation of a trauma-informed recovery planning approach:

**Recommendation 1. Understand historical context,** including: community history with traumatic events; other cultural/historic issues (e.g., racial or class tensions, land use, immigration, etc.); potential strains with government/authority; and, points of community strength and pride (Trauma-informed principle: *Cultural, historical, and gender issues*).

**Recommendation 2. Ensure access for all groups** by focusing outreach on historically vulnerable or marginalized populations, designing process to ensure access and equipping groups in decision-making processes to support empowerment and control (Trauma-informed principle: *Cultural, historical, and gender issues*).

**Recommendation 3. Review the physical site** of any client or community meetings and point out emergency exits, resources and procedures at the beginning of sessions (Trauma-informed principle: *Safety*).

**Recommendation 4. Acknowledge trauma and provide space for sharing**. Understand that some content may be triggering to the audience, have referrals for support and counseling and partner with mental health practitioners (Trauma-informed principle: *Safety*).

**Recommendation 5. Actively listen** and put community voices first; stay calm and do not become defensive (Trauma-informed principle: *Trustworthiness and transparency*).

**Recommendation 6. Support community partners** to facilitate communication and deliver messages. Be respectful of people’s time and compensate community-based organizations or other partners serving as intermediaries (Trauma-informed principle: *Trustworthiness and transparency*).

**Recommendation 7. Be candid about the actor’s role and capability**; do not overpromise. Do not try to be the hero—empower community heroes (Trauma-informed principle: *Trustworthiness and transparency*).

**Recommendation 8. Support peer-to-peer interactions by working** with a trained facilitator; designing stakeholder engagement for maximum participation and collaboration; allowing time for story sharing among participants; and providing links to resources (Trauma-informed principle: *Peer support*).

**Recommendation 9. Promote local resources** by partnering with local practitioners on projects, and with mental health providers and public health practitioners (Trauma-informed principle: *Peer support*).

**Recommendation 10. Support active leadership** by the local community to support ownership and control (Trauma-informed principle: *Empowerment and choice*).

**Recommendation 11. Provide meaningful alternatives** in an accessible language to enable community participation and direction. Allow time for local decision-making (Trauma-informed principle: *Empowerment and choice*).

**Recommendation 12. Support community decision-making and visioning for long-term recovery** by clarifying and disclosing how decisions will be made and how community input will be used. Present alternatives in simple and clear language. Lay out implications for short-, medium- and long-term decisions. Support community visions for long-term recovery (Trauma-informed principle: *Collaboration and mutuality*).

**Recommendation 13. Value community experience** by building in time for input and story-sharing, and honoring and seeking local knowledge and expertise (Trauma-informed principle: *Collaboration and mutuality*).

**Recommendation 14. Acknowledge that the actor does not have all the answers** and be clear about the practitioner’s role and scope. Engage people as active partners and participants (Trauma-informed principle: *Collaboration and mutuality*).

## 6. Implementation Considerations

Given the increasing frequency, cost and impact of disasters, the need for post-disaster recovery planning will only grow [8,27]. Those called in to plan for recovery are likely to have limited training in mental health and may lack awareness of the role trauma plays in communities following disasters; therefore, a trauma-informed approach can raise awareness of the role of trauma in the recovery context so that post-disaster recovery planning professionals can better support communities and individuals. 

This approach should not be seen as a call for those supporting post-disaster recovery planning to become mental health professionals or to provide services outside their training. Rather, the opportunity lies in helping these professionals recognize some of the forces driving behavior of community members and colleagues with whom they engage, and to recognize the value of working directly with local and public health and mental health partners that know their communities best. Through a trauma-informed approach and local partnerships, post-disaster recovery planning professionals can support more integrative and transformative approaches to disaster recovery, build trust, recognize historical and cultural sources of trauma, and address historical inequities. 

To operationalize a trauma-informed approach to recovery, planning, engineering and other professional accreditation organizations should offer resources to train recovery professionals. Training should focus on the awareness of a trauma-informed approach and how it aligns with core processes and expertise relevant to specific disciplines, as well as guidance on how to partner with public and mental health professionals. The goal is not to certify post-disaster practitioners to provide mental health services, but rather to help them understand mental health issues that may play a role in their work. Government agencies seeking external support for recovery planning and implementation should consider integrating trauma-informed language into requests for proposals (RFPs) for post-disaster technical assistance or other recovery services. Community engagement and long-range planning are often distinct and not integrated efforts, and in many cases are not supported at all through official recovery channels. We recommend the development of RFP templates with sample language for local governments to request that post-disaster recovery planning professionals who support their community leverage trauma-informed community engagement processes. Potential leaders for advancing these efforts include federal agencies (in particular, the Federal Emergency Management Agency (FEMA), National Oceanic and Atmospheric Administration (NOAA), and the Substance Abuse and Mental Health Services Administration (SAMHSA)), in partnership with professional associations (such as the American Planning Association, American Society for Adaptation Professionals and the American Society of Civil Engineers.)

This approach was developed based on our collective experiences working with communities following disasters and conducting interviews with additional practitioners in government and the private sector; however, further evaluation of this approach is required to optimize the processes and maximize the outputs and impacts. Moreover, although this approach was designed for use by practitioners engaging in longer range recovery efforts, additional stakeholders involved in more immediate recovery (i.e., those involved in helping community members meet basic needs) should be engaged to identify opportunities for adaptation for use in their own recovery-related activities, because they may benefit from using a similar approach.

Notably, although a trauma-informed approach to recovery aims to prevent re-traumatization, it may not prevent incidental physical or behavioral health impacts of recovery decisions. For instance, decisions on where to relocate healthcare services or grocery stores may hinder equitable access. Public health professionals should be engaged alongside post-disaster recovery planning professionals to consider the longer term health impacts of recovery decisions, using a more formal approach to health impact assessment [28]. 

## 7. Conclusions

A trauma-informed approach provides a pathway for post-disaster planning professionals to center community voices in recovery planning by raising awareness of the impact of trauma on the communities they serve and providing actionable strategies to acknowledge and address it. The 14 recommendations provided herein provide guidance for post-disaster recovery planning professionals supporting community recovery to build trust, engage residents and contribute to the development of healthy, resilient and sustainable communities after disasters.

## Figures and Tables

**Figure 1 ijerph-19-01723-f001:**
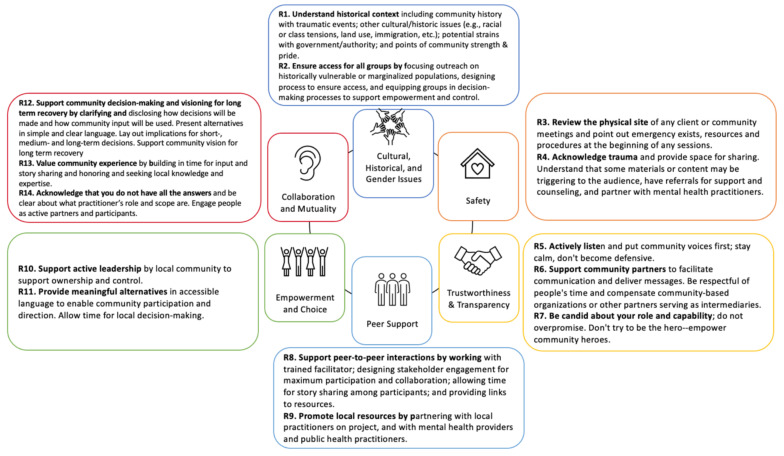
Recommendations for Trauma Informed Recovery Planning.

**Table 1 ijerph-19-01723-t001:** Trauma-informed recovery planning approach.

Principle	Definition Applied to Trauma-Informed Recovery Planning
Cultural, Historical and Gender Issues	Work with local partners to understand community context, sources of local pride and history of trauma. Ensure equitable access for all groups.
Safety	Support physical and psychological safety at all times.
Transparency and Trustworthiness	Build trust with the community through open, transparent communication, and be reliable and accountable for actions.
Peer Support	Provide space for peer-to-peer collaboration; promote local resources.
Empowerment and Choice	Recognize and empower community as decision-makers, and value community experience. Be reciprocal in interactions and avoid being extractive.
Collaboration and Mutuality	Promote community voices and initiatives. Provide options for participation and decision-making.

## Data Availability

Not applicable.

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
