# Peer review of "Working with Disaster-Affected Communities to Envision Healthier Futures: A Trauma-Informed Approach to Post-Disaster Recovery Planning"

_ijerph, 2022, doi:10.3390/ijerph19031723_

Round 1

Reviewer 1 Report

The paper is well-written and interesting.
Anyway, some suggestions for improvement should be considered, as follows:

- Line 34 pag.1: please add the reference to “build back better” (see UN Sendai Framework report).
- Line 125 pag.3: please replace “forums” with the correct Latin plural “fora”.
- Table 1 pag 4: please reduce the amount of words and make the table more readable, difficult to understand what belongs to what and where the rows are. Please notice that in the first column “safety” is double.
- Line 79 pag. 2 and line 211 pag. 5: the authors mention interviews and interviewers but no further information is given along the paper (i.e. how the interview was performed, people interviewed, questions asked, data collected, findings). Please contextualise it and provide more information on data, method and results obtained from the interview.

Author Response

Reviewer 1

The paper is well-written and interesting.

We thank the reviewer for their time and expertise, and appreciate the overall positive impression of our manuscript.

Anyway, some suggestions for improvement should be considered, as follows:

- Line 34 pag.1: please add the reference to “build back better” (see UN Sendai Framework report)

We have added a citation to the UNISDR’s 2017 Build Back Better in recovery, rehabilitation and reconstruction report.

- Line 125 pag.3: please replace “forums” with the correct Latin plural “fora”.

We have made this correction.

- Table 1 pag 4: please reduce the amount of words and make the table more readable, difficult to understand what belongs to what and where the rows are. Please notice that in the first column “safety” is double.

Thank you for this recommendation. We have integrated the recommendations into the body of the text, and created an additional figure aligning the trauma informed principles with our recommendations. Thank you for catching our error in the table; the  second “safety” has been replaced with “transparency and trustworthiness” as intended.

- Line 79 pag. 2 and line 211 pag. 5: the authors mention interviews and interviewers but no further information is given along the paper (i.e. how the interview was performed, people interviewed, questions asked, data collected, findings). Please contextualise it and provide more information on data, method and results obtained from the interview.

Thank you for this suggestion. We have added additional context as requested, and separated the interviews into a sub-section of the paper entitled “Challenges with post-disaster recovery planning.” Notably, these were formative interviews (not research interviews). We included some key sharings from interviewees in order to frame the paper and provide additional context and justification for a trauma-informed approach, not to imply these are results from any formal data collection and analysis process.

Reviewer 2 Report

line 36 to 39 should be clarified.

please revise and clarify this phrase" Following disasters, it is common for people to want to return to “normal” as soon as possible. However, as the impacts of climate change become more pronounced, previous conditions may no longer be fit for purpose..."

The proposed framework is not a framework!! it is like a table for summarizing some information. In addition, more elaboration is needed for this section.

In addition, the introduction is long and messy. for a reader, it is very hard to understand the logic of the introduction. and in some parts, I can see there is no consistency and the authors are talking about some offtopic matters. Please revise the introduction. this section should be more condensed.

In addition, the conclusion is very weak. it should be revised. 

Author Response

line 36 to 39 should be clarified.

Thank you for the opportunity to clarify. We have added that this is specific to the United States.

please revise and clarify this phrase" Following disasters, it is common for people to want to return to “normal” as soon as possible. However, as the impacts of climate change become more pronounced, previous conditions may no longer be fit for purpose..."

Thank you for the opportunity to clarify. In response to reviewer 1’s suggestion about providing additional information about the interview process, we have reworked this section. This text is now used to support an example provided by an interviewee, and has been revised and clarified to read: “​​This exemplifies the challenges of balancing community desires to quickly “return to normal” with the reality that previous conditions may no longer be able to withstand evolving hazard risks associated with a changing a climate.”

The proposed framework is not a framework!! it is like a table for summarizing some information. In addition, more elaboration is needed for this section.

We have augmented the description of the recommendations we provide as an “approach,” versus a framework. We have also added additional contextual framing in this section. Moreover, in response to Reviewer 1’s comments, we also moved the recommendations in the table to the main body of the text and to a descriptive figure.

In addition, the introduction is long and messy. for a reader, it is very hard to understand the logic of the introduction. and in some parts, I can see there is no consistency and the authors are talking about some offtopic matters. Please revise the introduction. this section should be more condensed.

Thank you for this suggestion. We have condensed the introduction and created a separate section on “Challenges to post-disaster recovery planning” that frames and discusses key sharings from interviews we conducted. We think this change has streamlined these sections and improved the logic and connection between the concepts introduced.

In addition, the conclusion is very weak. it should be revised.

Thank you for this suggestion. We have added more definitive language to the conclusion to strengthen it.

Round 2

Reviewer 1 Report

Fine as it is

Reviewer 2 Report

 Accept in present form